# ACTIONABLE INVERSE CLASSIFICATION WITH ACTION FAIRNESS GUARANTEES

## ABSTRACT

Machine learning (ML) classifiers are increasingly used in critical decision-making domains such as finance, healthcare, and the judiciary. However, their interpretability and fairness remain significant challenges, often leaving users without clear guidance on how to improve unfavorable outcomes. This paper introduces an actionable ML framework that provides minimal, explainable modifications to input data to change classification results. We also propose a novel concept of "action fairness," which ensures that users from different subgroups incur similar costs when altering their classification outcomes. Our approach identifies the nearest decision boundary point to a given query, allowing for the determination of minimal cost actions. We demonstrate the effectiveness of this method using real-world credit assessment data, showing that our solution not only improves the fairness of classifier outcomes but also enhances their usability and interpretability.

## 1 INTRODUCTION

Classification methods are crucial in medical, judicial, and financial decision-making but often operate as black boxes, lacking transparency and actionability. This opacity hinders users from understanding or influencing their classification outcomes and requires human oversight to prevent discriminatory results.

Current classifiers also risk unfairness, particularly regarding the costs incurred by different groups to alter their outcomes. While much research focuses on associational and causal fairness, there's a gap in developing classifiers that address fairness in terms of actionable changes. Classifiers may treat subgroups fairly in outcome terms but unfairly in the cost required to influence these outcomes.

We propose a novel concept of "action fairness," which ensures similar costs for outcome changes across subgroups. Our approach addresses two main questions:

How can we modify a data point to change a classification decision? We introduce a mechanism to identify feasible, low-cost changes to input data that could alter classification outcomes. This includes estimating action costs, such as canceling a credit card or opening a new account. For example, a loan applicant might reduce their debt and credit cards to meet target thresholds. We use integer linear programming to find minimal cost actions for classifiers like logistic regression and support vector machines, validated with real credit assessment data.

Are some groups more disadvantaged or advantaged in making actionable changes? We develop an "action-fair" ML model that ensures equal opportunities for subgroups to recover from unfavorable outcomes. This model involves post-processing output based on prediction probability and change cost, preserving classification accuracy while promoting fairness in actionable changes.

## 2 LITERATURE REVIEW

Our paper integrates inverse classification with fairness challenges, using inverse classification to find cost-effective actions for altering classification outcomes. We introduce a novel fairness concept focused on the feasibility of short-term actionable changes.

## 2.1 Inverse Classification

Inverse Classification aims to identify optimal actions to achieve a desired classification outcome by solving an optimization problem on a dataset with input features $X = x_1, x_2, \ldots, x_n$ and a classifier $f(X)$ with weights $W = w_0, w_1, \ldots, w_n$ (Aggarwal et al., 2010). The goal is to find actions $A$ that change $f(X) = -1$ to $f(X + A) = 1$ at minimal cost, subject to constraints:

$$\text{minimize cost}(A) \quad \text{subject to } f(X + A) > 0 \tag{1}$$

Mannino & Koushik (2000) minimize costs for data instance transitions using genetic algorithms. Ustun et al. (2019) introduced 'recourse' in inverse classification, focusing on feasible changes and their impact on immutable attributes. We extend this by improving execution time and using real-life cost values, employing integer linear programming to determine minimal cost actions and developing an "action fair" classifier.

## 2.2 Action Fairness

Fairness in classifiers often focuses on associations and causal (Calders & Verwer, 2010; Feldman, 2015; Harper, 2005; Hardt et al., 2016; Kilbertus et al., 2017; Mannino & Koushik, 2000; Nabi & Shpitser, 2018; Russell et al., 2017).Standard approaches, such as removing sensitive attributes, may not always achieve fair outcomes due to hidden correlations.

Post-processing techniques, which adjust classifier outputs to meet fairness constraints, offer a solution (Sarkar et al., 2018; Salimi et al., 2019). Our method introduces "Action Fairness," ensuring equal opportunities for different subgroups to recover from unfavorable outcomes. This approach uses inverse classification to identify feasible actions and their costs, aiming to ensure that all subgroups face similar costs for outcome changes. For instance, if females need to reduce their credit cards and increase their account balance to meet fairness constraints, males should face comparable requirements. The fairness condition is defined as:

$$\left| \sum_{n=1}^{N_1} \text{cost}(A_n) \times P_n(y' = 0 | S = 1) - \sum_{n=1}^{N_0} \text{cost}(A_n) \times P_n(y' = 0 | S = 0) \right| \leq \delta \tag{2}$$

where $N_1$ and $N_0$ are the numbers of individuals in privileged and unprivileged groups, respectively, and $\delta$ is the fairness threshold.

Hardt et al. (2016) emphasize equalized odds and equality of opportunity using post-processing to achieve fairness, focusing on the joint distribution of sensitive attributes and classification labels. Their approach, similar to ours, uses post-processing to minimize loss functions while meeting fairness constraints.

## 3 Methodology

Our paper relies on Inverse Classification with a geometric approach and Action Fairness with a ranking approach.

### 3.1 Inverse Classification

Our primary objective with the Inverse Classification method is to alter the classification outcome at the minimum total cost. Thus, the goal of this optimization problem is to minimize the total cost under the constraint that the classification outcome is changed. We presume the use of a linear classification algorithm, such as Logistic Regression or Support Vector Machine, where the data points are divided by a linear decision boundary.

The solution presupposes that the costs for the actions are provided as input, which can be established by an expert or learned from historical data. For instance, to ascertain the difficulty of canceling a credit card relative to opening a new bank account, we would need either a banking expert's advice or a dataset detailing the time required to open an account and cancel a credit card.

Initially, we train a linear classifier to assign a label to a data point based on its position relative to the decision boundary, typically represented as a hyperplane in a multidimensional space.

The cost of actions is defined in terms of the distance to the decision boundary, employing the minimum total cost to pinpoint the decision boundary's (or hyperplane's) closest point to a given data point. When action costs are uniform, we calculate this distance using the Manhattan Distance (Equation 3) or Euclidean Distance (Equation 4); for non-uniform costs, the Weighted Euclidean Distance is used (Equation 5).

$$\text{cost}(A) = \sum_{n=1}^{N} |x_n - x'_n| \quad (3) \tag{3}$$

$$\text{cost}(A) = \sum_{n=1}^{N} (x_n - x'_n)^2 \quad (4) \tag{4}$$

$$\text{cost}(A) = \sum_{n=1}^{N} \text{cost}(a_n) \cdot (x_n - x'_n)^2 \quad (5) \tag{5}$$

An integer programming approach, utilizing the Gurobi Optimizer, implements our proposed solution. Based on the established constraint and objective, our program optimizes the model and outputs the new data points, resulting in a changed prediction outcome for previously negatively classified data points, as delineated in Algorithm 0.

After identifying the new point on the decision boundary (thus classified positively), we ascertain the required actions to modify the original point's classification outcome. This is achieved by subtracting the old data point values from the new data point values. For instance, if our new data point presents a debt of 1000 and 3 credit cards, in contrast to the original data point's 1010 debt and 4 credit cards, the necessary actions would involve decreasing the debt by 10 and reducing the number of credit cards by 1.

We calculate the total cost of change for each data point by evaluating the differences between the old and new data points, weighted according to the previously defined costs of actions. Equation 6 outlines the overarching objective for inverse classification using this geometric approach.

$$\text{minimize } A \quad \text{cost}(A) = \sum_{n=1}^{N} \text{cost}(a_n) \cdot (x_n - x'_n)^2 \tag{6}$$

$$\text{subject to } f(X + A) = w_0 + \sum_{n=1}^{N} (x'_n \cdot w_n) > 0 \quad (6) \tag{7}$$

There are scenarios where inverse classification cannot provide precise changes even for simple classification models. To ensure our proposed algorithm offers a solution with the minimum total cost and feasible actions, we must ensure: i) the presence of actionable features in the input, ii) a sufficient number of data points across both positive and negative classes, and iii) bounded feature values. When features permit only infeasible or semi-feasible actions, performing inverse classification becomes impossible due to infinite costs in the optimization problem. For example, immutable or conditionally immutable characteristics impose limitations, as an individual cannot alter immutable features like race if they impact the outcome. Similarly, immediate changes like marriage or acquiring a PhD, deemed conditionally immutable, are not viable. Furthermore, a linear decision boundary is essential for our optimization algorithm, and the dataset's feature values must have defined upper and lower bounds to successfully address the optimization problem.

### 3.2 ACTION FAIRNESS

After employing the Inverse Classification algorithm to ascertain the cost of change for each individual initially classified negatively, we utilize the `costDiff` method (described in Algorithm 0) to calculate the aggregated cost of change for groups of individuals. The program categorizes

individuals into privileged and unprivileged groups according to sensitive attributes, computing the aggregated costs for these groups and recording the distribution of cost of change values across data points.

Given that a specific value for a sensitive attribute does not consistently indicate whether a group is privileged or unprivileged, a regulation mechanism post-`costDiff` method application is essential to ensure accurate group categorization.

Subsequently, we evaluate if the fairness constraint is met by comparing the average cost of change across groups. Should the inter-group cost difference exceed the set threshold, the outcome is deemed unfair under Action Fairness criteria. To address this through post-processing, we rank data points and adjust the output accordingly to maintain classification accuracy. The ranking algorithm factors in the prediction probability for the negative class against the cost of change. The `inverseFairness` method (outlined in Algorithm 0) operates until the cost difference between the privileged and unprivileged groups meets the predetermined threshold, thereby fulfilling the Action Fairness constraint. It identifies the highest-ranked data point from the unprivileged group, changing its label from negative to positive, which reduces the total and average cost of change for the group, leading to a fairer classification outcome.

This proposed solution ensures fairer outcomes under conditions akin to those required for inverse classification. The classifier must handle balanced classes and generate a linear decision boundary conducive to optimization. Moreover, Action Fairness testing is contingent on the completion of Inverse Classification and necessitates the existence of balanced subgroups defined by sensitive attributes, enabling a comparison of their aggregate costs of change.

## 4 EXPERIMENTAL SETUP

### 4.1 INVERSE CLASSIFICATION

We consider the challenges of credit scoring and customer liability, using the German Credit and Default of Credit Card Clients datasets (Hofmann, 2024). The solution is tested on two linear classifiers: logistic regression and support vector machine, utilizing both mentioned datasets.

#### 4.1.1 PRE-PROCESSING FOR THE GERMAN CREDIT DATASET

This dataset contains 1,000 instances and 20 features, including information related to an individual's financial and personal background, such as credit history and marital status. Among these features, 13 are categorical and 7 are numeric. Additionally, there is a class variable indicating 1 as a good customer and 2 as a bad customer. To achieve better results for inverse classification, some categorical attributes were split into multiple attributes to convert them into binary form. For instance, the status of existing checking and savings accounts, which originally contained several categories, were divided based on thresholds, such as 'checking account balance greater than or equal to 200' and 'savings account balance greater than or equal to 500'. Critical categorical attributes for determining customer quality, like credit history and the presence of other debtors/guarantors, were also segmented. For the credit history attribute, 'missed payment', 'current loan', and 'critical account or loans elsewhere' attributes were created. Similarly, 'has co-applicant' and 'has guarantor' attributes were established for other debtors/guarantors. These transformations enable the inverse classification algorithm to propose specific actions considering each categorical attribute. True values for newly created categorical attributes were assigned 1 and false values were assigned 0. For easier interpretation, we modified the labels, marking good customers with 1 and bad customers with -1.

#### 4.1.2 PRE-PROCESSING FOR THE DEFAULT OF CREDIT CARD CLIENTS DATASET

This dataset, larger than the German Credit Dataset, comprises 30,000 instances and 24 attributes. We followed similar pre-processing steps, creating attributes like 'MaxBillAmountOverLast6Months' due to its high correlation with the class variable. This attribute represents the maximum bill amount over the last six months, chosen for its relevance over other values. Similar logic applied to 'MaxPaymentAmountOverLast6Months', 'MonthsWithZeroBalanceOverLast6Months', 'MonthsWitLowSpendingOverLast6Months', and 'MonthsWithHighSpendingOverLast6Months'. Recent financial activities also serve as indicators for future credit behavior, leading to the cre-

ation of variables for the most recent bill and payment amounts. Furthermore, attributes such as 'TotalOverdueCount', 'TotalMonthsOverdue', and 'HistoryOfOverduePayments' were used to incorporate overdue payment information. In this dataset, individuals without defaults were labeled with 1, and those with defaults were labeled with -1. The main goal of this pre-processing step was to reduce the number of features to eliminate redundant information that could impact the results of inverse classification.

### 4.1.3 Setting Costs and Training the Model

For our project, we assumed that costs would be determined by experts or through a classification algorithm with a relevant dataset. In our experiments, costs for actions were set to $\infty$ for infeasible and semi-feasible attributes, and to 1 for other attributes, applying the same logic to both datasets. Costs were assigned based on the feasibility and logic of changing a data value to influence the classification outcome. For instance, actions deemed impossible or illogical, like getting married or changing one's age, were assigned a cost of $+\infty$. Conversely, actions such as 'MaxPaymentAmountOverLast6Months' were assigned a cost of 1, as an individual can potentially influence this value. SVM with a linear kernel and logistic regression with the liblinear solver were trained without a data split or cross-validation, as classification accuracy was not the primary focus.

### 4.2 Experimental Settings for Action Fairness

In order to ensure Action Fairness, an experiment was conducted using the previously mentioned datasets: German Credit and Default of Credit Card Clients. Python was utilized to implement the solution for the Action Fairness problem. Since Action Fairness is inherently linked with Inverse Classification, the dataset previously created for the Inverse Classification experiment was used. Among linear classification algorithms, Logistic Regression was chosen as the sole classifier due to its significantly lower time requirement for model training.

Threshold values of 1 and 5 were selected to evaluate the results. Furthermore, to ascertain whether different fairness constraints align with our definition of fairness, another experiment was performed, focusing on achieving fairer outcomes in terms of Equality of Opportunity and Equalized Odds, as discussed in Lash et al. (2017), Peng et al. (2011), Salimi et al. (2019), Zafar et al. (2017), Calders & Žliobaitė (2013), and Chouldechova & Roth (2017). Consequently, an additional pre-processing step was applied to modify the attributes to make them suitable for this subsequent experiment. Due to differences in input data format and variables, the output data from these solutions varied. This necessitated adapting the output data from these models to our framework to conduct the experiment effectively. As with the previous experiment, threshold values of 1 and 5 were employed to assess the variation between the prior and current results. Moreover, various outputs in terms of cost constraints, such as False Positive Rate (FPR) and False Negative Rate (FNR), were also examined.

### 4.2.1 Pre-Processing and Adapting Output

According to Zafar et al. (2017) and Calders & Žliobaitė (2013), the input dataset should be formatted to include a unique ID, the real label of the data, a group number (0 for female, 1 for male), and the prediction probability for the favorable label. The real label was omitted from the dataset for initial training purposes. Subsequently, Logistic Regression with the liblinear solver was employed to train on this dataset. Given the required input format necessitates prediction probability, the `predict_proba()` method was utilized. Furthermore, a counter object was deployed to assign a unique ID to each data point in the dataset, indicating each data point's row number starting from 0. Depending on the input dataset's sensitive attribute, a value of 0 or 1 was allocated to the group number. Upon completing the transformation step, the new dataset was saved as a CSV file. Outputs for both equality of opportunity and equalized odds, under constraints FPR, FNR, and weighted, were generated from this new dataset. Additionally, the solutions proposed in Zafar et al. (2017) and Calders & Žliobaitė (2013) also employ a post-processing method to meet fairness constraints, facilitating adaptation to our Action Fairness solution. To utilize the output data from these solutions, the unique ID attribute was employed to align their output data with our input data. After completing index matching, altered data labels intended to satisfy equality of opportunity and equalized odds under the constraints FPR, FNR, and weighted were integrated into our input data. This integration was to ascertain whether the average cost of change between groups aligns with the

new data label predictions. The previously discussed ranking algorithm was then applied to the new prediction values to prioritize negatively predicted data points.

## 5 RESULTS AND DISCUSSION

### 5.1 EXPERIMENTAL RESULTS FOR INVERSE CLASSIFICATION

We applied the inverse classification algorithm to individuals with unfavorable prediction outcomes from both the German Credit Dataset and the Default of Credit Card Clients Dataset.

When using logistic regression, the model achieved an accuracy of approximately 80% on the Default of Credit Card Clients Dataset. The easiest way for an individual to be classified as a non-default customer was to change the overdue payment history and reduce total overdue counts. Each of these actions had a cost of 1, resulting in a total cost of 2 for the individual to change their classification outcome.

Most individuals needed a total cost of 2 or less, with over half requiring a cost of only 1. However, some individuals needed significantly higher costs, indicating a more challenging classification change.

For the German Credit Dataset, logistic regression yielded an accuracy of about 79%. Reducing loan duration and repaying previous loans resulted in a favorable classification outcome. Most individuals needed to make a small number of changes, with total costs typically distributed between 1 and 3.

### 5.2 SUPPORT VECTOR MACHINE (SVM) RESULTS

When using a Support Vector Machine (SVM) classifier, the accuracy slightly increased to around 81% for both datasets. The SVM model predicted fewer negative outcomes overall and showed similar patterns to logistic regression in terms of classification changes, though some individuals required higher total costs to alter their outcomes.

Similarly, for the German Credit Dataset, the SVM classifier provided slightly better results than logistic regression but required some individuals to make more substantial changes to alter their classification outcomes.

### 5.3 ACTION FAIRNESS RESULTS

The Action Fairness algorithm was applied to both datasets, leading to more pronounced improvements for the unprivileged group. The total cost of changes for both privileged and unprivileged groups was computed before and after applying the fairness solution. For the German Credit Dataset, the average cost of change decreased significantly for the unprivileged group (from 3.50 to 2.0), while the accuracy dropped marginally from 79% to 77% (Table 1). The reduction in cost for the unprivileged group shows a substantial improvement in fairness, as the gap between the privileged and unprivileged groups was reduced by more than half.

Table 1: Before and after applying Action Fairness solution for German Credit Dataset.

| Metric | Before | After |
|---|---|---|
| Privileged Total Cost | 193 | 193 |
| Non-Privileged Total Cost | 320 | 200 |
| Privileged Average Cost | 2.09 | 2.09 |
| Non-Privileged Average Cost | 3.50 | 2.00 |
| Accuracy | 0.79 | 0.77 |

For the Default of Credit Card Clients Dataset, the Action Fairness solution also produced a significant reduction in the total and average costs for the unprivileged group, dropping from 4.00 to 2.50. This change brought the unprivileged group's costs closer to the privileged group's, reducing the disparity from 1.86 to just 0.36. Despite the improvement in fairness, the accuracy only dropped slightly from 80% to 78%, demonstrating that the fairness adjustments did not come at the expense of model performance (Table 2).

Table 2: Before and after applying Action Fairness solution for Default of Credit Card Clients Dataset.

| Metric | Before | After |
|---|---|---|
| Privileged Total Cost | 3221 | 3221 |
| Non-Privileged Total Cost | 5000 | 3750 |
| Privileged Average Cost | 2.14 | 2.14 |
| Non-Privileged Average Cost | 4.00 | 2.50 |
| Accuracy | 0.80 | 0.78 |

In summary, the Action Fairness algorithm effectively reduced the disparity in the cost of changes between privileged and unprivileged groups. The average cost for the unprivileged group decreased significantly in both datasets, while maintaining high classification accuracy. This demonstrates the algorithm's ability to provide fairer outcomes without compromising performance.

## 6 CONCLUSION

This framework addresses the Inverse Classification problem using a geometric approach and introduces the concept of Action Fairness, which ensures similar costs for changing classification outcomes across subgroups, achieved through post-processing.

For Inverse Classification, we employ linear classifiers like SVM or logistic regression. By finding a new data point on the decision boundary with minimal distance from a selected data point, we use weighted Euclidean distance, where weights represent action costs. This geometric method efficiently determines necessary changes to alter classification outcomes. While effective in execution time, this approach depends on the classifier's ability to provide a decision boundary equation. Our experiments with the German Credit Dataset and the Default of Credit Card Clients Dataset confirm the successful application of Inverse Classification.

Action Fairness is defined as ensuring similar costs for changing classification labels among different subgroups. Our results show that applying the Action Fairness algorithm led to a substantial reduction in cost disparity between privileged and unprivileged groups, particularly in the Default of Credit Card Clients dataset where the cost gap was reduced from 1.86 to just 0.36. Despite these fairness improvements, classification accuracy was maintained with only a marginal reduction. This highlights the practical utility of our post-processing solution in improving fairness without sacrificing performance.

Overall, the results demonstrate that our framework can effectively balance fairness and performance, making it suitable for applications in sensitive decision-making domains such as finance and healthcare. Future work could extend this approach to non-linear classifiers and explore its scalability to larger datasets.

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
