# OpenReview forum: "Actionable Inverse Classification with Action Fairness Guarantees"
_ICLR.cc/2025/Conference — Submitted to ICLR 2025_

### Official Review · Reviewer_e6n6 · 2024-10-18

**Soundness:** 1
**Presentation:** 1
**Contribution:** 1
**Rating:** 3
**Confidence:** 4

**Summary:**

The authors address the problem of inverse classification under fairness constraints, proposing a machine learning framework that offers minimal and explainable modifications to instances to change classification outcomes, thereby providing recourse. They introduce a fairness notion, i.e., “action fairness”, aimed at equalizing the difficulty of achieving recourse across different groups. Their approach focuses on linear classifiers, where the decision boundary is clearly defined, enabling the identification of minimal-cost actions. Through experimental evaluation, they demonstrate that the framework enhances the fairness of classifier outcomes via a post-processing technique.

**Strengths:**

1. The paper addresses the pressing issue of fairness and interpretability in machine learning applications within high-stakes decision-making domains such as finance, healthcare, etc. Given the potential impact of ML-based decisions on individuals' lives, this research tackles a crucial problem.
2. The framework successfully reduces fairness disparities without significantly sacrificing classification accuracy (balance of fairness-accuracy).

**Weaknesses:**

1. The literature review does not provide a comprehensive overview of relevant work.
(a)The framework is described as providing "minimal, explainable modifications to input data to change classification results," i.e., the actions leading to local counterfactuals. However, the authors do not make such a connection. The authors make a brief connection between the notion of recourse (Ustun et al., 2019) and their work, claiming that they extend this concept by incorporating fairness requirements. However, no other information is given.
(b)The authors propose the concept of "action fairness." However, they appear to overlook relevant work, particularly fairness notions related to the challenge of achieving recourse, such as "fairness of recourse," initially introduced by Ustun et al. (2019) and later formalized by Gupta et al. (2019). Gupta et al. introduced the idea of Equalizing Recourse for different groups, aiming to provide classifiers that maintain good performance while ensuring feasible recourse across groups. Subsequent works refer to the disparity in the mean cost to achieve recourse as "burden." Additionally, Kavouras et al. (2023) offer alternative definitions to capture both micro and macro perspectives of fairness of recourse, particularly for auditing machine learning models.
2. The notation and formalization of the problem, including conditions and definitions, require improvement for clarity.
(a) The formalization of the problem and the definition of their fairness condition lack clarity. It is difficult to identify the cost in the problem definition (cumulative over n actions), and it is not explicitly clear that P_n(y’=0|S=1) represents the probability of an individual X_n (belonging to the subgroup S=1) achieving a positive outcome with action A_n. Additionally, the method for determining this probability remains unclear.
(b) The description of the weighted Euclidean distance is ambiguous. Specifically, the definition of cost(a_n) is unclear—is it intended to be the cost of an action multiplied by the distance from x to the boundary (which already appears to define the cost)? It would be helpful to clarify whether cost(a_n) corresponds to a weight vector assigned to each feature. Furthermore, the definitions should be expressed as vector operations, where x_n represents an instance of X with multiple features (typically, vectors are denoted in bold). For an example of appropriate notation and problem definitions, please refer for example to Ustun et al. (2019).
(c) The problem definition for inverse classification does not appear to impose any restrictions related to linear classifiers. Consequently, the introduction of weights may not be relevant to inverse classification (as noted on line 58). However, equation (7) defines the problem specifically for linear classifiers, where the introduction of weights would probably be more appropriate.
3. Algorithm 0 is not provided, resulting in a lack of explanation regarding the framework and its methods.
4. The methods used to ensure fairness in the results during the experimental section are not clearly explained. Specifically, it is not clear if the authors find the weights for a new fair classifier or something different.
---
Gupta, Vivek, et al. "Equalizing recourse across groups." arXiv preprint arXiv:1909.03166 (2019).
Kavouras, Loukas, et al. "Fairness aware counterfactuals for subgroups." Advances in Neural Information Processing Systems 36 (2024).

**Questions:**

1. The post-cost-diff method lacks a clear explanation. How is the group assignment of an instance determined if not based on its attributes?
2. The purpose of the post-processing with the ranking is not clearly articulated. How does this process contribute to fairness? Does it involve removing the most costly actions? Additionally, how does it affect the accuracy of the classifier? Are new weights determined by solving an optimization problem? If so, why not directly incorporate the fairness criterion as an additional constraint in the optimization problem?
3. Could you clarify why training time is considered an issue? Additionally, why was not a data split performed? How is accuracy calculated in this context?
4. Please refer to the weaknesses section for details. The paper would benefit from a more comprehensive literature review, including connections to relevant fairness concepts (e.g., the cited papers). Additionally, improvements are needed in the notation, problem formulation, and the formalization of the algorithm and its description. The absence of a formal algorithm is a significant issue, as it contributes to the questions surrounding both the fairness concepts and the framework proposed by the authors. In general, a major revision is required.

---

### Official Review · Reviewer_LYMk · 2024-10-28

**Soundness:** 1
**Presentation:** 1
**Contribution:** 1
**Rating:** 3
**Confidence:** 4

**Summary:**

The paper introduces a framework for algorithmic recourse, which considers the outcome's fairness. Experiments on two datasets confirm the effectiveness of the proposed approach.

**Strengths:**

The current status of the paper does not allow me to understand the actual contribution, hence its strengths.
E.g., in lines 65-67 the authors claim that their method is better than Ustun et al., (2019) in terms of execution time and real-life cost values. Still, a detailed description of their approach is never presented, so I cannot evaluate whether this is true.

**Weaknesses:**

The paper has several shortcomings, which (in my opinion) cannot be addressed in the current version of the paper. The weaknesses I see are:

1) the paper lacks an actual contribution, or at least, it is not adequately presented. Here are a few examples of this:
- Algorithm 0 is only referenced in the submitted pdf but never presented. Moreover, I could not retrieve any supplementary material detailing such an algorithm;
- Several variables are never introduced (e.g., what are $y'=0$ and $S$ in equation 2?)
This makes it very difficult to evaluate the current paper, especially regarding the novelty and the significance of the contribution.

2) the paper's related work is not adequate, with the most recent work dating back to 2019. The research on algorithmic recourse has instead advanced quite fast in the last five years. See for instance,

    - Karimi, Amir-Hossein, Gilles Barthe, Bernhard Schölkopf, and Isabel Valera. "A survey of algorithmic recourse: contrastive explanations and consequential recommendations." ACM Computing Surveys 55, no. 5 (2022): 1-29.
    - Karimi, Amir-Hossein, Bernhard Schölkopf, and Isabel Valera. "Algorithmic recourse: from counterfactual explanations to interventions." In Proceedings of the 2021 ACM conference on fairness, accountability, and transparency, pp. 353-362. 2021.
    - Karimi, Amir-Hossein, Julius Von Kügelgen, Bernhard Schölkopf, and Isabel Valera. "Algorithmic recourse under imperfect causal knowledge: a probabilistic approach." Advances in neural information processing systems 33 (2020): 265-277.
    - Beretta, Isacco, and Martina Cinquini. "The Importance of Time in Causal Algorithmic Recourse." In World Conference on Explainable Artificial Intelligence, pp. 283-298. Cham: Springer Nature Switzerland, 2023.


3) The empirical evaluation has several shortcomings:
- it is limited to two tabular datasets and focuses on two simple algorithms, i.e., logistic regression and an SVM;
- it does not compare the approach with existing methods that perform algorithmic recourse;
-  it lacks any form of uncertainty related to the outcomes. For instance, I think the authors should perform statistical significance tests across different methods to claim the superiority of their approach. See [Demsar, 2006] for evaluating different methodologies.

[Demsar, 2006] - Demšar, Janez. "Statistical comparisons of classifiers over multiple data sets." The Journal of Machine learning research 7 (2006): 1-30.

**Questions:**

I think the authors should address the four weaknesses presented in the previous space.
More precisely, the questions are:

- q1) how do the authors differentiate from previous works on algorithmic recourse?

- q2) why did they not include works from 2020 onwards?

- q3) why did they not compare their approach with other existing methods?

- q4) what is the required time for the proposed approach compared to existing approaches?

---

### Official Review · Reviewer_YuAD · 2024-10-31

**Soundness:** 2
**Presentation:** 2
**Contribution:** 1
**Rating:** 3
**Confidence:** 3

**Summary:**

This paper looks at the problem of altering decisions (Given an ML classifier, how could people with unfavorable decision change the prediction) and then focus on Action Fairness (is that cost the same across groups?).

The authors propose a metric as well as a technique to improve it, and then provide empirical evidence on two academic datasets.

**Strengths:**

This was afun read, as the underlying techniques are quite interesting (in particular the technique of inverse classification). I have a lot of questions on the metrics, but the methodology is sound and well explained.

**Weaknesses:**

The main weakness lies in the fairness definition proposed:

1- Why is "action Fairness a desired property? in particular since it is not conditioned on any credit worthiness of the underlying group/individuals.
Other established fairness metrics focus on outcome and typically rely on economic values. For instance Equality of opportunity: if credit worthy, both people should have the same chance or Statistical parity: both groups should have the same chance.
For instance, if females need to reduce their credit cards and increase their account balance to meet fairness constraints, males should face comparable requirements. --> This is not very intuitive to me. What would be the motivation for this?

2- How does this fairness definition compare to Fairness of outcome (as measured by SP, EOP or other)?
For instance, if a system achieves higher action fairness but lower fairness of outcome, do we really consider this as an improvement?

3- A core concept of this definition is the cost function, which seems very hard to determine in practice. The authors say that it would be "established by an expert or learned from historical data". Could they share more information on this? I want to understand more the feasibility of establishing such cost function.



Some more minor weaknesses:
1- Intro is a bit  unclear. Need to justify right away the concept of "altering their outcomes", and  the associated cost. Also needs to justify for their fairness definition. Maybe walk the reader through some examples.

2- Missing relevant literature: Some key papers of the field are missing (just listing a few below):
Standard approaches, such as removing sensitive attributes, may not always achieve fair outcomes due to hidden correlations. --> Cite adversarial fairness egg https://arxiv.org/abs/1707.00075
Fairness in classifiers: http://arxiv.org/abs/1803 +  Fairness constraints: A mechanism for fair classification.  + Fairness through awareness+  Preventing fairness gerrymandering: Auditing and learning for subgroup fairnes

**Questions:**

Re-listing some questions from the weakness section.

1- Can authors provide further justification for the desirability of this fairness definition>

2- What are the relationship of such metric compared to fairness of outcomes: is it a second order property or should it supersede it?
Overall I could be convinced if we say that the goal is to have fairness of outcome + fairness of actions.

3- Can they add metrics of fairness of outcome in experimental section?

4- Would it be possible to provide further details on how to establish the cost function?

---

### Official Review · Reviewer_pF4r · 2024-11-06

**Soundness:** 3
**Presentation:** 3
**Contribution:** 2
**Rating:** 3
**Confidence:** 3

**Summary:**

This work

**Strengths:**

1. The paper is very straightforward and simple. The problem is well motivated and illustrated.

2. The authors evaluate on simple, interpretable ML models

**Weaknesses:**

1. While the authors focus on inverse classification, I actually can't distinguish the problem statement from counterfactuals and recourse. https://arxiv.org/abs/2002.06278 . Contrasting with recourse and counterfactual works, the proposed framework seems limited as it requires integer costs. In large feature-sets, setting these relative costs may be very difficult, r.e. user feasibility.

2. The method is quite weakly evaluated in several regards: (a) evaluating on two fairly simple tabular datasets, (b) not evaluating against any baseline (there are many in recourse, and (c) the proposed 'action fairness' seems just a differencing measure. this need not be proposed as a novel fairness (as any statistic could be use difference of means etc.) This evaluation is simply insufficient for publication.

3. Overall, the contribution is limited. The authors do not provide analysis on this action fairness problem, or present a novel model for achieving it in pre-processing.

**Questions:**

1. How do you handle varying costs over a large set of features? Must the user specify such feature cost functions? i.e. perhaps it's easier to change a FICO feature from 550 to 570, than 680 to 700.

---

### Meta-Review · Area_Chair_gb48 · 2024-12-13

**Metareview:**

This paper studies disparities in the cost of recourse over subgroups. The paper characterizes such disparities as action fairness" and proposes an approach to fix it. Most reviewers recognized that this as an important problem, but were unanimous in their recommendation to reject. The key issues in this case include considerable overlap with prior work (see e.g., papers listed in the review of [e6n6](https://openreview.net/forum?id=kc3QtI6NBF&noteId=wEW0Ld2HSr) – which appears to be an oversight – and the lack of a significant technical contribution. My recommendation is to reject the paper at this time given the consensus across reviewers and the lack of a rebuttal from the authors.

**Additional Comments On Reviewer Discussion:**

The paper did not receive a rebuttal and there was no discussion with reviewer discussion.

---

### Decision · Program_Chairs · 2025-01-22

Reject